# Ecosystem Service Valuation along Landscape Transformation in Central Ethiopia

**Abera Assefa Biratu** [1,2,*], **Bobe Bedadi** [1], **Solomon Gebreyohannis Gebrehiwot** [3], **Assefa M. Melesse** [4], **Tilahun Hordofa Nebi** [2], **Wuletawu Abera** [5], **Lulseged Tamene** [5] and **Anthony Egeru** [6,7]

1 Africa Centre of Excellence for Climate Smart Agriculture and Biodiversity Conservation, Haramaya University, Dire Dawa P.O. Box 138, Ethiopia; bbedadi@haramaya.edu.et
2 Melkassa Agricultural Research Center, Ethiopian Institute of Agricultural Research, Adama P.O. Box 436, Ethiopia; tilahun_hordofa@yahoo.com
3 Ethiopian Institute of Water Resource, Water and Land Resource Center, Addis Ababa University, Addis Ababa P.O. Box 3880, Ethiopia; solomon.g@wlrc-eth.org
4 Department of Earth and Environment, Institute of Environment, Florida International University, Miami, FL 33199, USA; melessea@fiu.edu
5 International Center for Tropical Agriculture (CIAT), Addis Ababa P.O. Box 5689, Ethiopia; wuletawu.abera@cgiar.org (W.A.); lt.desta@cgiar.org (L.T.)
6 Department of Environmental Management, Makerere University, Kampala P.O. Box 7062, Uganda; a.egeru@ruforum.org
7 Regional Universities Forum for Capacity Building in Agriculture (RUFORUM), Kampala P.O. Box 16811, Uganda
* Correspondence: abera.assefa@eiar.gov.et

**Abstract:** Land degradation and discontinuation of ecosystem services (ES) are a common phenomenon that causes socio-economic and environmental problems in Ethiopia. However, a dearth of information is known about how ES are changing from the past to the future with regard to land use land cover (LULC) changes. This study aimed at estimating the values of ES based on the past and future LULC changes in central Ethiopia. Maximum likelihood classifier and cellular automata-artificial neuron network (CA-ANN) models that integrate the module for land use change evaluation (MOLUSE) were used to classify and predict LULC. The CA-ANN model learning and validation was employed to predict LULC of 2031 and 2051. Following LULC change detection and prediction, the total ES values were estimated using the benefit transfer method. Results revealed that forests, wetlands, grazing lands, shrub-bush-woodlands, and water bodies were reduced by 9755 ha (37%), 4092 ha (38.4%), 21,263 ha (81%), 63,161 ha (25.7%), and 905 ha (1%), respectively, between 1986 and 2021. Similarly, forests, wetlands, grazing lands, shrub-bush lands, and water bodies will experience a decline of 1.5%, 0.5%, 2.6%, 19.6%, and 0.1%, respectively. Meanwhile, cultivated lands, bare-lands, and built-up areas will experience an increase between 1986 and 2051. The estimated total ES values were reduced by US$58.3 and 85.4 million in the period 1986–2021 and 1986–2051. Food production and biological control value increased while 15 other ES decreased throughout the study periods. Proper land use policy with strategic actions, including enforcement laws for natural ecosystems protection, afforestation, ecosystems restoration, and conservation practices, are recommended to be undertaken to enhance multiple ES provision.

**Keywords:** landscape transitions; ecosystem services; ecosystem service valuation; CA-ANN; MOLUSE

## 1. Introduction

Ecosystem services (ES) impairment has been an important problem affecting livelihoods and human well-being [1]. Biodiversity loss and land degradation as well as climate change have been indicated as the major drivers of ES impairments [2,3]. While land degradation and biodiversity losses can be triggered by several factors, land use/cover (LULC) conversions have been identified as the most prominent factors in space and time [4,5].

LULC change entails the conversion of physical entities/cover type (e.g., forest to culti-vated land, or grazing land to cultivated land) and the form people utilize the land (e.g., rain-fed to irrigation agriculture) [6]. Conversions of natural ecosystems (i.e., natural forest, rangeland, wetlands) to modified/artificial ecosystems (i.e., agro-ecosystem, urban areas) are some of the most prominent and reported processes in the landscape transitions [7]. However, LULC changes have inherent variations and are non-linear in space and time [4,8]. Consequently, varied patterns and extent and form of change can be observed in each LULC type [9,10] due to the variation of the drivers. Multidimensional interconnections of human-induced drivers, such as socio-economic and institutional policies together with environmental factors, continue to influence LULC across the landscapes [11,12]. Moreover, the ever-growing demand for tillable land and infrastructure as populations continues to rise exerts more pressure on land resources and accelerating LULC [13].

Ethiopia is no exception to accelerated LULC that are anthropogenically induced with considerable effects on land and livelihoods [5]. For example, it is estimated that about 1.5 billion tons of topsoil is eroded per annum from the Ethiopian highlands [13,14]; soil fertility of the country has decreased by 122, 13, and 82 kg ha$^{-1}$ yr$^{-1}$ of N, P, and K, respectively. [15]. Meanwhile, crop production has declined by 32% [12,15]; and forest cover of the country has been reducing by 140 thousand ha$^{-1}$ yr$^{-1}$ (on average 1.0–1.5%) since 1990 [16]. These dynamics have led to an estimated overall environmental damage cost estimated to be about \$4.6 billion yr$^{-1}$ [14]. Within Ethiopia, the Central Rift Valley (CRV) is immensely affected. Previous studies [17–21] show that severe soil erosion originates from the untreated upper parts of agricultural landscape and sediment loading into the streams affecting Lake Zeway, Lake Abjiyata, and Lake Langano with nutrients (phosphate, nitrate, and silicate). It has also been estimated by Mukai et al. [22] that soil erosion in the area reached up to 140 t h$^{-1}$ yr$^{-1}$, while Aga et al. [23] have shown that sediment yield into lake Zeway was about 2.081 Mt yr$^{-1}$. These processes have led to reduced water quality and increased eutrophication challenges [24,25]. Transitions of natural ecosystems has often led to diminished ES provision capacity of the landscapes [26]. These are, in part, the consequence of LULC trade-offs associated with policies in favor of agricultural production and expansion [27,28]. Though efforts have been made to curb land degradation, there remains a dearth of information on the dynamics of landscape transformations and associated ES value.

Therefore, LULC-based ecosystem services valuation (ESV) is paramount to unravel the spatially explicit monetary value of ES and their changes in the landscapes [29–31]. It is a critical aspect required to guide landscape management policies and decisions undertaken by policy and decision leaders [29,32,33]. The ESV is relatively easy to apply because of the application of proxy data (existing ESV data) through the benefit transfer method, where the environmental hazard is prevalent with limited data [30,34]. Owing to the high cost of performing the original valuation, the application of benefit transfer valuation method provides an affordable and viable alternative to estimate ES value [29]. It is a prompt and cost effective approach to inform and catch more decision maker's attention on environmental hazards and associated costs [35,36]. As a result, ESV has been the widely used and emerging approach in contemporary ecosystem studies. For instance, emerging modeling tools used for ESV are 'Artificial Intelligence for Ecosystem Services' ARIES [37]; Co\$ting Nature v.3 'C\$N' [38]; 'Integrated Valuation of Ecosystem Services and Tradeoffs' InVEST [39]; 'Multiscale Integrated Models of Ecosystem Services' MIMES [40]; and 'Social Values for Ecosystem Services' SolVES [41]. ESV is paramount to grasp information about marketable and non-marketable ES bundles in the ever changing conditions and the possible land management options that enhance ES provision.

Landscape scale studies with regards to the past and future LULC transitions that trigger the loss and degradation of natural ecosystems would be crucial to curb and reverse the loss and assure sustainability in central Ethiopia. Studies to explore and prioritize degraded areas prone to ES impairment and loss would also be important to identify meticulous prompt decisions that help to comprehend and mobilize appropriate

interventions. In this study, we assessed, predicted, and analyzed LULC transitions and associated changes on ES value in the Rift Valley Basin of Ethiopia.

## 2. Materials and Methods

### 2.1. Study Area Descriptions

The Rift Valley Basin is located between 38°15′ E and 39°25′ E longitude and 7°10′ N and 8°30′ N latitude covering the surface area of about 10,074 Km² (Figure 1). Its altitude ranges between 1646 masl nearby Lake Zeway and 4171 masl on the ridges of Chilalo Mountain. It is a hydrologically closed basin and has three lakes (Zeway, Langano, and Abjiyata) and four perennial rivers (Ketar, Meki, Horakela, and Bulbula). The estimated mean annual inflow contributions of Ketar is 406 million m³ and of Meki is 260 million m³ to Lake Zeway [42]. Lake Abjiyata is connected with Lake Zeway through Bulbula River and has annual streamflow of 200 million m³. The landform of the study area has three distinct regions; rift, escarpment, and highland. The rift floor region has semi-arid climate conditions while the highlands experience humid and sub-humid climatic conditions. Mean annual precipitation is 900 mm yr$^{-1}$ with a variable distribution range between 700 and 200 mm per annum. On the other hand, mean annual temperature is around 15 °C in the highlands and 20 °C in the rift floor.

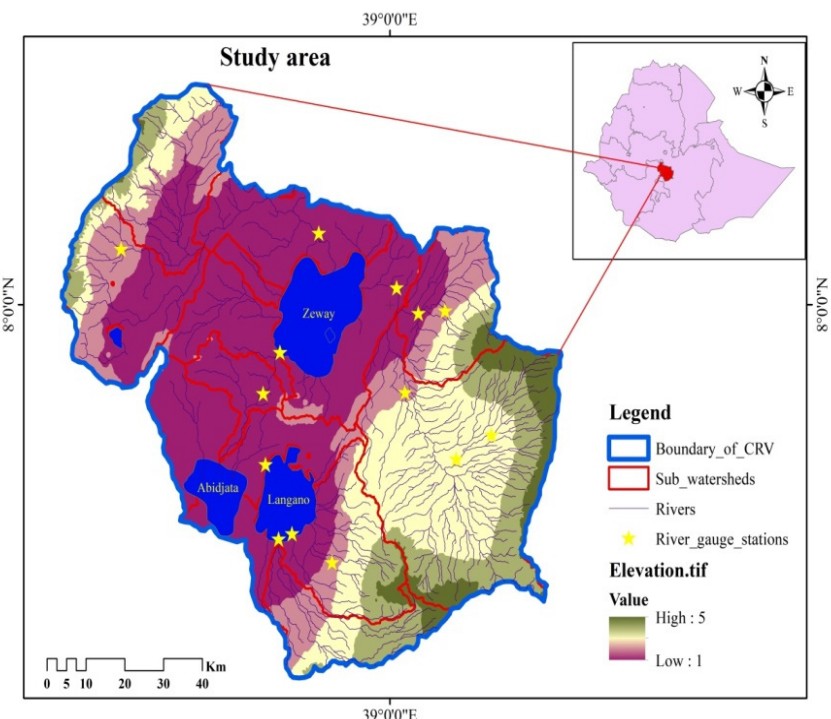

**Figure 1.** Map of the study area of the Central Rift Valley in Ethiopia.

Dominant soil types in the study area include Vertisols (Mazic Vertisols/Aridic Haplusterts) in the rift floor, Calcaric Fluvisols in the Meki River delta, and Eutric Nitisols on the plateaus of the western and eastern margin [43]. The soils found on the mountain and major escarpments are generally well-drained, very deep, and vary from dark reddish-brown to dark brown, and clay loam to loamy soils [42]. Soils with moderate infiltration dominate the rift floor around Lake Zeway, while the slow infiltration rate soils dominate at the escarpments [43]. The major crops produced under rain-fed agriculture include maize, faba bean, wheat, teff, and barley.

### 2.2. Data

Different types of data were acquired from various sources. Four time series Landsat satellite images and digital elevation model (DEM) with 30 m resolution were obtained

from United States Geological Survey (USGS https://earthexplorer.usgs.gov/ (accessed on 1 March 2022)). Landsat of Multispectral Scanner (MSS), Thematic Mapper (TM), Enhanced Thematic Mapper Plus (ETM+), and Operational Land Imager (OLI) were obtained for 1986, 2001/2011, and 2021 study periods, respectively. These cloud free Landsat images were acquired in January and February. The path and raw of these Landsat data were 169/54&55. Important landscape features, such as roads, river networks, population density of 2016, contour, built-up, as well as Topo-map (1:100,000) were obtained from the EthioGIS MapServer Ethiopia web site (www.mapserver-ethiopia.org). All spatial data were prepared to the same reference systems (WGS_1984_UTM_Zone_37N) and then cropped to fit the study landscape in order to use and proceed for further process. Distance from road, river, and market were prepared using DEM and corresponding features in ArcGIS environment. Similarly, altitude and slope classes were prepared using contour and DEM. Ground trothing data and information was undertaken in 2018 and 2021 using participatory field observation, interviews, and GPS ground control points. Google Earth imagery of corresponding study periods was also used as an ancillary information provider.

### 2.3. LULC Classification, Accuracy Assessment and Change Detection

Model Builder of ArcGIS 10.3 software was used to make composite bands and mosaicked images of 1986, 2001, 2011, and 2021. Geo-referencing and rectification was done to adjust distortion using ground points and roads network using ArcGIS. Following geo-correction, atmospheric corrections were performed to remove the haze and nose from images using ERDAS Imagine 15 software. Images were then resampled using the study area boundary. Unsupervised image classification was employed for each corresponding study period in order to use for ground assessment during participatory observation and interview with local elders to record oral history. Then, a supervised classification procedure was conducted for image classification. The training points that were proportionally distributed to each cover type were taken based on field observations, GPS ground points, and Google Earth images. Maximum likelihood classifier was used in a supervised classification procedure to classify the images independently in ERDAS Imagine 15 software. In order to assess the accuracy of image classification, crucial post classification evaluation was done through the confusion matrix method using testing data sets. The pattern and transitions of LULC classes that are described in Table 1 were determined and detected over studied years. The percent and rate of changes were also calculated to elucidate observed LULC transitions.

### 2.4. Modeling and Prediction of LULC

In the contemporary LULC studies, modeling and prediction become an emerged and broadly applied for ecological and environmental analysis [44]. According to IPCC [44] in LULC predictions there are two types of scenarios (i.e., exploratory and normative) used to unfold the future LULC. An exploratory scenario is a prediction of future LULC changes based on extrapolation of past trends and reference for present day conditions used as inputs. A normative scenario is a prediction of LULC based on a predetermined future that enables the use of references to consider every aspect (i.e., worst and optimal cases) or to explore extreme events.

In this study a module for land use change evaluation (MOLUSCE) plug-in for QGIS 2.0 was used to model and predict future LULC transitions using artificial neural networks (ANN). ANN is a perceptron model that developed as a simplified mathematical method. The simplified features of the ANN were simulated based on complex biological neurons interconnections that create neural networks. It is a non-deterministic model with some level confidence on the outputs. ANN uses independent variables as input to predict target outputs. According to IPCC, Ref [44] using selected input variables as a baseline (or reference) that allows for addressing the present day socio-economic, institutional, and environmental conditions is highly recommended to predict the future LULC.

**Table 1.** Description of LULC classes.

| LULC | Descriptions |
|---|---|
| Cultivated land | Area under rain-fed and/or irrigation cultivations |
| Forest | Area covered by plantation as well as natural forest trees |
| Water | Permanently covered by water (i.e., lakes, reservoirs, rivers) |
| Grazing land | Area covered with grass that regularly under animal browse |
| Wetland | Area covered by long grasses and shrubs that tolerant for waterlogged conditions |
| Built-ups | Area covered by built-ups wherein dense population live |
| Bare land | Area with no vegetation, mainly exposed rocky and sand covered area |
| Shrub-bush land | Land covered with shrubs and thorn bushe plants, Afro-alpine vegetation and/or acacia tree with undergrowth grasses |

We used slope, elevation, distance from road, distance from market, distance from river, rainfall, and population density as independent variable inputs to train ANN and compute nonlinear function for simulation. These independent variables used in this study were identified based on previous LULC studies (e.g., [33,44–47]). We also used 2001, 2011, and 2021 LULC maps for ANN model training and validation in order to predict 2031 and 2051. Simulations were employed in ten year time interval and then 2031 and 2051 were selected as optimum years that prompt for the mid and long term prediction. Based on the study area characteristics, current situation, and development plan, we assumed similar trends for the next four decades to undertake simulation.

All identified input variables were prepared in raster with similar geometry and analysed using Pearson's correlation method in MOLUSCE. Henceforth, transition potential modeling was employed following customization of the multi-layer perceptron to train the ANN model. Accordingly, the training process of the model was performed with 100 iterations, 3 × 3 neighborhood value, 0.001 learning rate, 10 hidden layer, and 0.050 momentum values based on Bugday et al. [48], and then the neural network learning curve graph was produced. Accordingly, the transition probability matrix was computed using quantitative analysis to the classified LULC maps.

Following the transition potential calculation and training processes, validation and simulation were performed. The cellular automata-artificial neuron network (CA-ANN) model was performed under three processes (ANN, CA, and validation) that enabled the model conveniently to simulate and produce LULC maps. The model follows the Markovian approach to compute the transition probabilities matrix using the cellular automata-artificial neuron network (CA-ANN) model. The validation process was performed using the classified 2021 LULC and simulated 2021 LULC. The validation result on the accuracy of a simulated LULC map was checked based on kappa statistics. After a while, simulations were conducted using the CA-ANN model. Then, LULC area changes, gain, and loss for respective study years were calculated.

*2.5. Ecosystem Services Valuation (ESV)*

We employed ESV based on a value transfer approach of the benefit transfer method that uses ES coefficients values. It is known that in the absence of site specific valuation information, the benefit transfer method is an alternative to perform ESV. Two ways of beneficial transfer approaches are available: function transfer and value transfer. Function transfer approach predicts value coefficients for the new study site based on the available data, whereas value transfer is an approach used to transfer the overall value from the original site to the new site. The benefit transfer method was used to perform ESV over the study period 1986–2051. We used ES value coefficients modified by Kindu et al. [49] for the Munessa-Shashemene area, which is similar to the study landscape (Table 2). Coefficients were modified through a benefit transfer method based on expert knowledge of the study area conditions, a previous study of Costanza et al., [31] and a valuation database of the Economics of Ecosystems and Biodiversity (TEEB). In addition, values were mainly

extracted from the TEEB database of tropical areas of LULC types that fit the study area geographical setting.

**Table 2.** Modified ES value coefficients (million US$ ha$^{-1}$ yr$^{-1}$) for LULC equivalents land cover.

| LULC | Equivalent Land Cover | ES Coefficient (Million US$ ha$^{-1}$ yr$^{-1}$) |
| --- | --- | --- |
| Cultivated land | Cropland | 225.56 |
| Forest | Tropical forest | 986.69 |
| Water | Lakes | 8103.5 |
| Grazing land | Grass/Rangelands | 293.25 |
| Wetland | Swamps/Floodplains | 8103.5 |
| Built-ups | Urban | 0 |
| Bare land | Desert | 0 |
| Shrub-bush land | Grass/rangelands | 293.25 |

ESV was calculated using the following steps and equations [49].

$$ESV_{Kt} = \sum (A_K \times VC_K) \tag{1}$$

where; ESV = total estimated ES value, $A_k$= the area (ha) and $VC_k$ = the value coefficient (US$ ha$^{-1}$ year$^{-1}$) for LULC type 'k'.

The total ESV for reference year 't' calculated by adding the ESV of each LULC class as follows

$$ESV_t = \sum (ESV_{Kt}) \tag{2}$$

The change in ESV over time was assessed using the formula:
Percent of ESV change =

$$[(ESV_{t2} - ESV_{t1})/ESV_{t1} \times 100] \tag{3}$$

Thus, the total change in ES value was estimated by calculating the differences between the estimated values for corresponding LULC classes. The individual ecosystem functions in the study landscape was calculated by using the following equation

$$ESV_f = \sum (A_K \times VC_{fK}) \tag{4}$$

where; $ESV_f$ = calculated ES value of function 'f', $A_k$ = the area (ha) and $VC_{fk}$ = value coefficient of function 'f' (US$ ha$^{-1}$ year$^{-1}$) for LULC type 'k'.

The modified coefficients values for 17 individual ecosystem services are presented in the second figure in Section 3.3, and were also obtained from [49]. The contributions of individual ecosystem functions to the overall value of ES per year were ranked based on an estimated value of ecosystem functions for each reference year. The percentage change was determined in $ESV_f$ for a given value coefficient following the standard economic method. Principal component analysis (PCA) of LULC and different ES was employed to analyze the relationship between LULC and ES, and also in identifying the trade-off and synergies observed between ES in the landscape using R software version 4.0.

## 3. Results

### 3.1. Land Use and Land Cover (LULC) Changes

LULC compositions, extent, and changes of the study landscape (CRV) in the study period (1986–2021) are presented in Table 3 and Figure 2. The results of this analysis revealed an overall average classification accuracy of 90.7% with an average Kappa index of 0.83 for all of the years of analysis.

**Table 3.** LULC changes in the study periods (1986 to 2021).

| LULC | LULC Area in (ha) | | | | LULC Area Changes in (%) between Study Periods | | | | Gain and Loss in (%) Based on the LULC Total Area Coverage |
|---|---|---|---|---|---|---|---|---|---|
| | 1986 | 2001 | 2011 | 2021 | 1986–2001 | 2001–2011 | 2011–2021 | 1986–2021 | 1986–2021 |
| Cultivated land | 593,175 | 659,050 | 660,734 | 666,489 | 11.1 | 0.3 | 0.9 | 12.4 | 7.3 |
| Forest | 26,432 | 18,923 | 18,633 | 16,677 | −28.4 | −1.5 | −10.5 | −36.9 | −1.0 |
| Water body | 82,489 | 80,671 | 78,757 | 81,584 | −2.2 | −2.4 | 3.6 | −1.1 | −0.1 |
| Grazing land | 26,230 | 4421 | 5690 | 4967 | −83.1 | 28.7 | −12.7 | −81.1 | −2.1 |
| Wetland | 10,664 | 9273 | 8614 | 6572 | −13.0 | −7.1 | −23.7 | −38.4 | −0.4 |
| Built-ups | 1462 | 4192 | 5143 | 7880 | 186.8 | 22.7 | 53.2 | 439.2 | 0.6 |
| Bare land | 21,502 | 27,829 | 31,889 | 40,945 | 29.4 | 14.6 | 28.4 | 90.4 | 1.9 |
| Shrub-bush | 245,398 | 202,993 | 197,890 | 182,237 | −17.3 | −2.5 | −7.9 | −25.7 | −6.3 |
| Total | 1,007,351 | 1,007,351 | 1,007,351 | 1,007,351 | | | | | 0 |

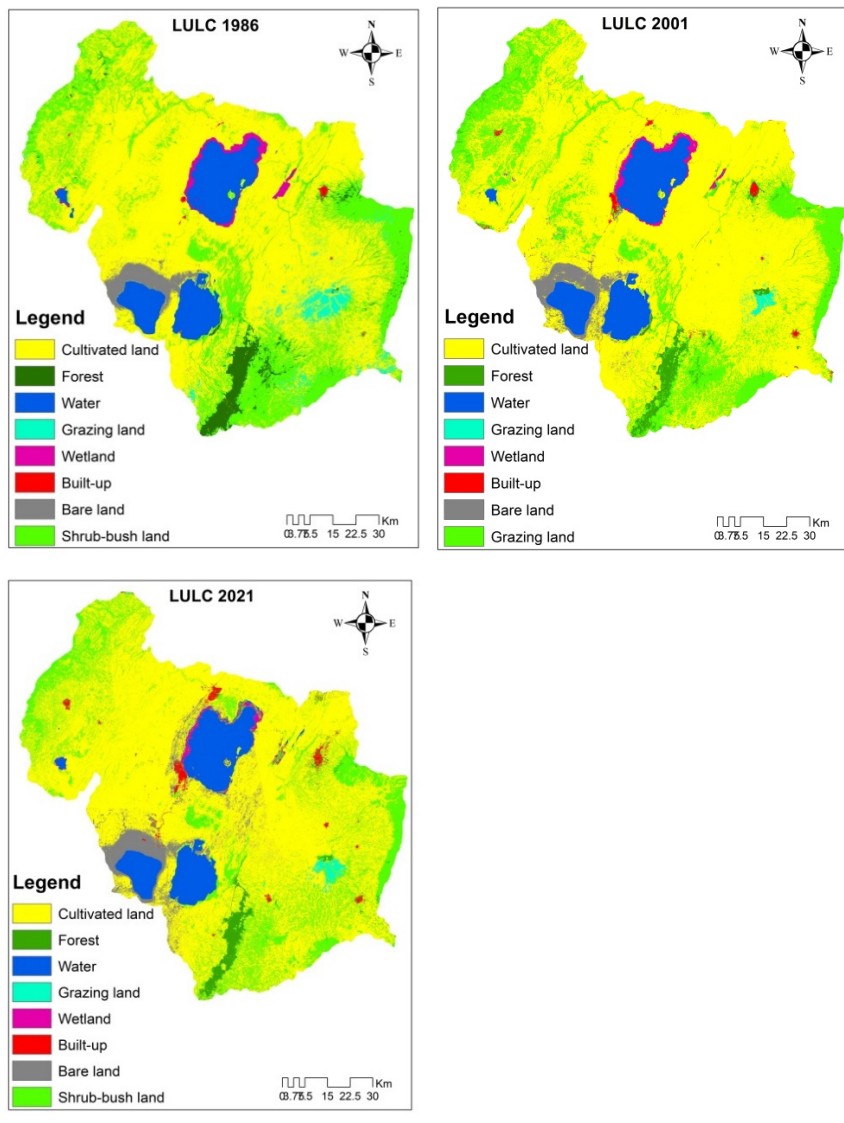

**Figure 2.** LULC class of study landscape in order, 1986, 2001, and 2021.

Of all LULC cultivated land, bare land and built-up areas were immensely and persistently increased in the year between 1986 and 2021. On the contrary, forest, grazing land, wetland, and shrub-bush-woodland were significantly decreased. However, water bodies

were reduced with minimum extent compared to other LULC. Shrub-bush land, wetland, and forest were persistently reduced throughout the study periods (1986–2021) (Table 3).

Across all LULC, the highest change was detected on cultivated land by 73,315 ha between 1986 and 2021 (Table 3). On the other hand, the least change was detected on water by 905 ha difference. The substantial percent increments were observed for built-up areas by 187% and 303.4% between 2011–2021 and 1986–2021, respectively.

We found that LULC in the area was variable across the study period. From 1986, cultivated lands, shrub-bush lands, and water bodies were found with higher area coverage throughout the study period. Contrarily, forests, wetland, bare, and built-up areas were with the least area coverage. Area coverage of all LULC at the end of the study period, 2021, was not similar to the initial study period, 1986.

### 3.2. Predicted LULC Changes 2031–2051

The predicted LULC area and maps of the study landscape for 2031 and 2051 are shown in Table 4 and Figure 3. The validation of results shows 80% of correctness, and a robust Kappa (histo) 0.95020. Forest, water, grazing land, wetland, and shrub-bush land have significant area loss in between the study periods of (1986–2031) and (1986–2051). Cultivated land, built-up areas and bare land are predicted to continue to gain more area in the years to come (2031 and 2050). Grazing land will become a lost habitat by 2051 (Figure 3). Transition matrix results (Table A1) revealed varying probabilities of change within and across land covers for the period of analysis (1986–2021, 2021–2031, and 2021–2051).

**Table 4.** Predicted LULC area (ha) and the gain and loss in (%) based on the LULC total area coverage across the study periods.

| LULC | Predicted LULC Area in (ha) | | Gain and Loss in (%) Based on the LULC Total Area Coverage between Periods | | | |
|---|---|---|---|---|---|---|
| | 2031 | 2051 | 2021–2031 | 2021–2051 | 1986–2031 | 1986–2051 |
| Cultivated land | 795,733 | 813,037 | 12.8 | 14.5 | 20.1 | 21.8 |
| Forest | 12,072 | 11,089 | −0.5 | −0.6 | −1.4 | −1.5 |
| Water body | 81,513 | 80,998 | 0.0 | −0.1 | −0.1 | −0.1 |
| Grazing land | 1927 | 159 | −0.3 | −0.5 | −2.4 | −2.6 |
| Wetland | 5875 | 5461 | −0.1 | −0.1 | −0.5 | −0.5 |
| Built-ups | 6242 | 5260 | −0.2 | −0.3 | 0.5 | 0.4 |
| Bare land | 25,868 | 43,592 | −1.5 | 0.3 | 0.4 | 2.2 |
| Shrub-bush | 78,122 | 47,755 | −10.3 | −13.4 | −16.6 | −19.6 |
| Total | 1,007,351 | 1,007,351 | 0 | 0 | 0 | 0 |

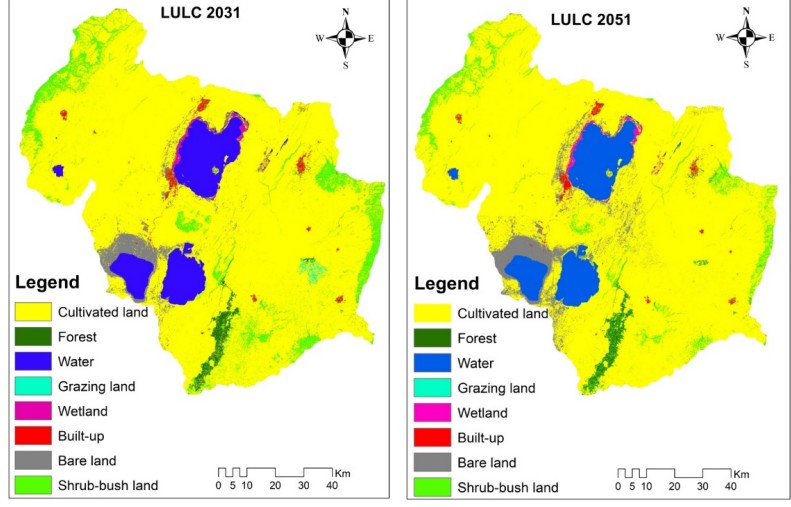

**Figure 3.** LULC maps of predicted 2031 and 2051.

With the exception of cultivated land and bare land, all simulated LULC classes will experience a decline in 2031 and 2051. Drastic transitions of LULC were observed in the periods 2021–2031, 2021–2051. The changes in the most predicted LULC have a similar trend to the observed LULC. Cultivated land and bare land will increase by 2051 as all the other land uses and covers continue to decrease by 2051.

The loss and gains of LULC were higher in the natural ecosystems than modified ecosystems.

### 3.3. Ecosystem Services Valuation (ESV) to Historical and Future LULC

The estimated ESV of the study landscape were US$994.4, 957, 951.7, and 936.1 million in 1986, 2001, 2011, and 2021, respectively (Table 5). The total ES value was persistently reduced by US$37.4, 5.3, and 15.7 million during the study periods 1986−2001, 2001−2011, and 2011−2021, respectively (Table 5).

**Table 5.** ESV (US $ million ha$^{-1}$yr$^{-1}$) of each LULC class and its change over study periods (1986 to 2021). Figures in parentheses indicate a change of ESV in percent (%).

| LULC Class | ESV US$ Million ha$^{-1}$ yr$^{-1}$ | | | | Change of ESV in US$ Million ha$^{-1}$ yr$^{-1}$ and (%) between Period | | | |
|---|---|---|---|---|---|---|---|---|
| | 1986 | 2001 | 2011 | 2021 | 1986–2001 | 2001–2011 | 2011–2021 | 1986–2021 |
| Cultivated land | 133.8 | 148.7 | 149.0 | 150.3 | 14.9 (11.1) | 0.4 (0.3) | 10.5 (7.5) | 16.5 (12.4) |
| Natural forest | 26.1 | 18.7 | 18.4 | 16.5 | −7.4 (−28.4) | −0.3 (−1.5) | −19.6 (−54.4) | −9.6 (−36.9) |
| Water body | 668.4 | 653.7 | 638.2 | 661.1 | −14.7 (−2.2) | −15.5 (−2.4) | 22.3 (3.5) | −7.3 (−1.1) |
| Grazing land | 7.7 | 1.3 | 1.7 | 1.5 | −6.4 (−83.1) | 0.4 (28.7) | −6.0 (−80.5) | −6.2 (−81.1) |
| Wetland area | 86.4 | 75.1 | 69.8 | 53.3 | −11.3 (−13) | −5.3 (−7.1) | −16.6 (−23.7) | −33.2 (−38.4) |
| Built-ups | 0.0 | 0.0 | 0.0 | 0.0 | 0.0 | 0.0 | 0.0 | 0.0 |
| Bare land | 0.0 | 0.0 | 0.0 | 0.0 | 0.0 | 0.0 | 0.0 | 0.0 |
| Shrub-bush land | 72.0 | 59.5 | 57.0 | 53.4 | −12.4 (−17.3) | −2.5 (−4.2) | −3.6 (−6.3) | −18.5 (−25.7) |
| TOTAL | 994.4 | 957.0 | 951.7 | 936.1 | −37.4 (−3.8) | −5.3 (−0.6) | −15.7 (−1.6) | −58.8 (−5.9) |

The ESV of the study area was reduced by US$58.8 million in the year between 1986 and 2021 (Table 5). Cultivated land was the only land use that increased ESV by US$16.5 million (12.4%) in the years between 1986 and 2021. The cultivated land related ESV increased by 11% and 7.5% in the years from 1986−2001 and 2011−2021. The ES value of grassland, forest, wetland area, water body, and shrub-bush land reduced by 81.1, 36.9, 38.4, 1.1, and 25.7%, respectively, in the years between 1986 and 2021. The largest reduction of ESV was US$33.2 in the years from 1986−2021 associated with the loss of wetland. Transitions of shrub-bush land in 1986−2021 and forest in 2011−2021 resulted in substantial ESV reduction by US$18.5 and 19.6 million, respectively. Among all LULC, wetland and shrub-bush land related ESV reduced persistently and drastically.

Total ESV will be reduced by US$71.4 million and US$85.4 million in the study year 1986−2031 and 1986−2051, respectively. Similarly, ESV will be reduced by US$13 and 27.1 million in the study years from 2021−2031 and 2021−2051, respectively, in the study landscape (Table 6). In the predicted year 2051 grazing land related ESV would be nil. Of all LULC, shrub-bush land related ES values would be immensely reduced by US$49.1 and 58 million in the years 1986−2031 and 1986−2051, respectively. On the contrary, cultivated land related ES value will increase by US$45.7 and 49.6 million in the years 1986−2031 and 1986−2051, respectively. In the next three decades, wetlands and shrub-bush lands account for US$42.1 and 58 million ESV reductions, respectively. However, cultivated lands will see an increase in ESV by US$3.9 million (19%) by 2031. Overall, ESV related to shrub-bush land, wetland, grazing land, and water will reduce in the next one to three decades.

**Table 6.** ESV (US$ million ha$^{-1}$yr$^{-1}$) of each LULC class and its change over study periods (1986 to 2051). Numbers in parentheses indicate a change of ESV in percent (%).

| LULC | ESV in US$ Million ha$^{-1}$ yr$^{-1}$ | | Changes in ESV in US$ Million ha$^{-1}$ yr$^{-1}$ between Study Periods | | | |
|---|---|---|---|---|---|---|
| | **2031** | **2051** | **2021−2031** | **2021−2051** | **1986−2031** | **1986−2051** |
| Cultivated land | 179.49 | 183.4 | 3.9 (19) | 29.2 (22) | 45.7 (19) | 49.6 (37) |
| Forest | 11.91 | 10.9 | −1.0 (−28) | −4.5 (−34) | −14.2 (−28) | −15.2 (−58) |
| Water body | 660.54 | 656.4 | −4.2 | −0.6 (−1) | −7.86 | −12 (−2) |
| Grazing land | 0.56 | 0.0 | −0.5 (−63) | −0.9 (−100) | −7.1 (−63) | −7.7 (−100) |
| Wetland | 47.61 | 44.3 | −3.4 (−11) | −5.6 (−17) | −38.8 (−11) | −42.1 (−49) |
| Built-ups | 0.00 | 0.0 | 0.0 | 0 | 0 | 0 |
| Bare land | 0.00 | 0.0 | 0.0 | 0 | 0 | 0 |
| Shrub-bush land | 22.9 | 14.0 | −30.5 (−57.1) | −39.4 (−73.8) | −49.1 (−68.2) | −58 (−80.5) |
| Total | 923.0 | 909.0 | −13 (−8) | −27.1 (−11) | −71.4 (−8) | −85.4 (−19) |

Of all estimated 17 ecosystem services (ES) presented in Figure 4, food production and biological control will increase by 7.6 and 3.1% in 2031, and 7.2 and 1.5% in 2051, respectively. In the years between 1986 and 2021, only food production experienced an increase in ESV$_f$ while the rest of ES reduced.

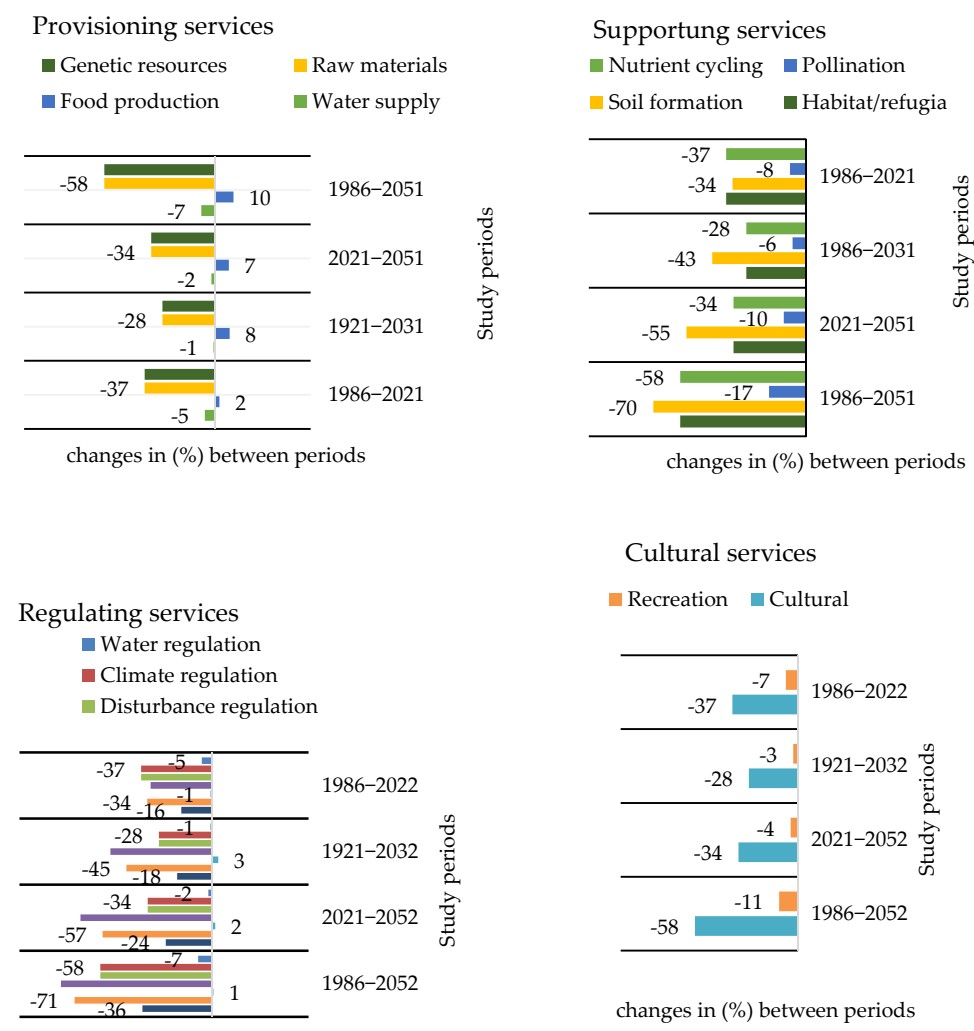

**Figure 4.** Individual ecosystem service (ESVf) changes in percent (%) between the study periods.

Figure 5 shows the PCA results that present the relationships between LULC and ES as well as between individual ES. Across all LULC classes, shrub-bush land has high positive value for gas regulation, erosion control, soil formation, raw materials, climate regulation, habitat/refuge, nutrient cycling, genetic resources, and cultural and disturbance regulation (ES are sorted from the strongest) (Figure 5a). This reveals a high contribution of shrub-bush land to gas regulation, erosion control, soil formation, raw materials, climate regulation, habitat/refuge, nutrient cycling, genetic resources, and cultural and disturbance regulation compared to other LULC classes in the study landscape. Shrub-bush land has a wide area coverage that enables it to influence important ES compared to other natural ecosystems in the study landscape. Water bodies have strong positive relationships with water supply, water regulation, and recreation. The contribution of water bodies to enhance water supply, water regulation, and recreation services was very high. It also has no negative impact to increase biological control, food production, and pollination services. Cultivated land has a high value for biological control, food production, and pollination, but a negative correlation waste treatment. Besides cultivated land, the contribution of forest and grazing land was positive, but not significant, to enhance biological control, food production, and pollination. The influence of LULC on ES depends on their area coverage in the study landscape. For instance, in this study, the contribution of forest and grazing land to influence the total ES values was not significantly detected in the PCA.

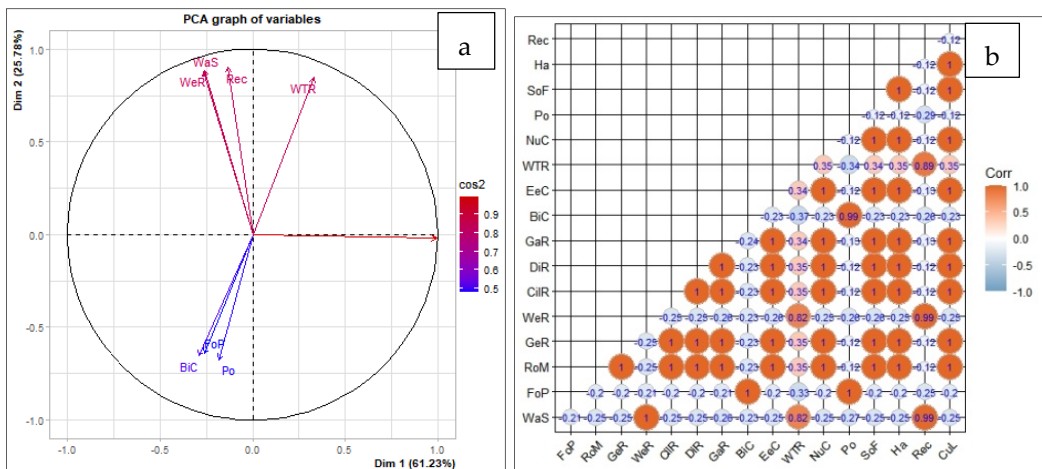

**Figure 5.** Principal component analysis (PCA) of LULC and different ESs. *Note*: (**a**) is an ES (variables) factor map that is labeled ES; those are selected shown on the plane; (**b**) is a graph that shows the correlations between ESs. Each ES were denoted as follows: Water supply (WaS); Food production (FoP); Raw materials (RoM); Genetic resources (GeR) Water regulation (WeR); Climate regulation (CilR); Disturbance regulation (DiR); Gas regulation (GaR); Biological control (BiC); Erosion control (EeC); Water treatment (WTR); Nutrient cycling; (NuC); Pollination (Po); Soil formation (SoF); Habitat/refuge (Ha); Recreation (Rec); Cultural (CuL).

The trade-offs and synergies between ES were identified based on correlation analysis result of PCA (Figure 5b). The ES, such as raw materials, genetic resources, climate regulation, disturbance regulation, gas regulation, erosion control, nutrient cycling, soil formation, habitat/refuge, and cultural, have a positive correlation. ES, such as water supply, water regulation, and recreation have positive correlations. Only water supply and water regulation have marginal correlations with the rest of the 15 ES. Food production, biological control, and pollination have a positive correlation between each other, but negatively correlate with the rest of the 14 ES. Waste treatment has negative correlations with food production and biological control, but positively correlated with the rest of the 15 ES. These negative correlations identified between ES shows a trade-off of ES that needs to be addressed.

## 4. Discussion

In this study, we analyzed historical (1986–2021) and future (2031–2051) LULC and associated changes in order to assess their impact on ES value in the landscapes in Ethiopia. The results showed accelerated landscape change with an observed reduction in natural ecosystems, especially wetlands, grazing lands, and forests, while cultivated, built-up areas and barren lands increased overtime (Table 3). These patterns could be attributed to competition for agricultural and settlement land. Previous studies conducted in Ethiopia (e.g., [6,33,50]) reported similar results. It is mainly due to a lack of proper land use policies and population growth associated demands of cultivation and settlement land. Unlike in this study, some studies reported that the dynamism of forest area is not predictable and tough to define its changes over time. Among several factors, regime change related policy changes were an important factor to alter and make forest area coverage unpredictable [51,52]. Changes triggered by the interplay of several drivers were identified related to social, economic, environmental, policy/institutional, and technological factors [11].

Cultivated land, bare land, and built-up areas increased throughout the study period (Table 4). This is mainly to satisfy the demand of the growing population to produce food, to secure shelter, and expand industries and factories. Excessive pressure on, and improper uses of, resources due to growing population coupled with climate changes causes natural ecosystems to convert to modified ecosystems. In addition, sustainable agricultural intensification and ecosystem restoration were not properly implemented to restrict further expansion of cultivated land. The most prominent anthropogenic drivers for LULC changes are population growth, expansion of cultivated lands and settlements, livestock ranching, cutting of woody species for fuelwood, and charcoal making [11]. However, these anthropogenic drivers are location specific and needed to develop intervention strategy for sustainable development. This result is consistent with previous studies reported with different percent increments [6,53]. Bush-shrub land and grass land are important ecosystems that contracted through time and commonly converted to cultivated and bare land (Table 5 and Figure 3). This is consistent with [34] study, which revealed significant reduction of grass and bush-shrub-woodland in the Afar region. The transitions trend shows that grassing land ecosystems habitat could disappeared and be replaced by others in the near future in the landscape. The aquatic ecosystems, such as lakes and wetlands were also substantially reduced (Table 3). Similar results were reported in previous studies (e.g., [54–56]). However, it is not consistent with the findings reported that waterbodies increased by 38,427 ha during 1986–2017 in the Afar region [34]. There are several anthropogenic and environmental factors have been contributed to water bodies' reduction. Of all factors, improper water uptake from lakes and rivers for irrigation purpose, increasing domestic water consumptions, and climate changes were the major causes that risks Rift Valley lakes and aquatic ecosystems [57–59]. For instance, Lake Abidjata has been at the edge of extinction and a few small fragmented wetlands were converted to cultivated land [28]. As a result, aquatic and terrestrial species diversity that depends on the lakes and wetland ecosystems have been considerably threatened. This substantial loss of biodiversity is expected to affect ES, livelihoods, and human well-being in the landscape. The influences of population growth and socio-economic activities that result in expansion of large scale investment, arable lands, and infrastructures coupled with inefficient land use policies enforcement were the root cause for observed transitions [60]. As a result, the structures and functions of landscape ecosystems remain disturbed and inefficient to provide services.

Therefore, the implication of the LULC changes on ESV were significant and estimated to reduce by US$85.4 million in 2050 compared to 1986 in the study landscape. Several previous studies conducted in different parts of Ethiopia also reported substantial ESV reduction related to LULC changes. For instance, Shiferaw et al. [61] in Gojeb Omo-Gibe; Shiferaw et al. [34] in Afara; Tolessa et al. [62] in Dendi; Aneseye et al. [63] in Winike Omo-Gibe; and Tolessa et al. [10] in Toke-Kutaye reported ESV reduction by US$551, 112, 47, 36.4, and 36.3 million, respectively, in the years between 1973−2018. This is

contrary with [64–66] studies that were done in different locations of Ethiopia. Studies that used different coefficients revealed similar ESV reduction trends in similar study periods (e.g., [49,57]). The magnitude and rate of LULC changes varies in different locations. It is due to non-linear properties of LULC changes, land management differences, and high variations in landscape characteristics that could be the reason for ESV discrepancies. The maximum exhibited ESV reduction was US$1091 million in the Afar region by and the maximum increment was US$345 million at the Abaya-chamo basin, Ethiopia. However, studies have been performed with some apprehensions raised on ESV chorus. For instance, uncertainties, double counting, and underestimation of the monetary values that regarded some LULC classes [29]. Toman [64] explain ESV as a suspicious methodology that follows reductionistic computation. Though economic valuation is not preferential to apply to the entire range of biodiversity benefits, it is effective at the ecosystems level and has a significant contribution for social value [65,66]. Despite the apprehensions ESV can still be safely used and sound to provide immediate information about landscape ecosystems change, which is complex and non-linear.

Trade-offs analysis among the whole ES related to LULC changes is paramount for proper land use policy development that assures sustainability. The correlation analysis identified the trade-offs (i.e., negative correlation among ES) and synergies (i.e., positive correlation among ES) observed among ES related LULC changes (Figure 4). Important challenges that were observed in relation to LULC changes were the trade-offs between agricultural food productions and the rest of the important ES (Figures 4 and 5). This is due to conversions of natural ecosystems to cultivated land in order to satisfy food demands of the growing population. As a result, sustainability of agricultural production and environment were significantly threatened. It is paramount to manage the trade-offs observed between ES through suitable land use that restrict further agricultural land expansion and practice sustainable intensifications in the landscape. The loss of ESV observed in different parts of Ethiopia due to LULC changes implies that provisions of ES and human well-being are threatened. The land use policies and enforcement strategies need to be revised in order to transform the landscapes for ES provisioning enhancement and sustainability.

*Implications for Sustainable Landscape Management*

This study showed that there were significant landscape transformations between the study periods 1986–2021 and will continue drastically in the coming three decades. As a result, ES impairments and a significant reduction of ES values will be observed. This implies that several inhabitants of the study landscape in particular and the country in general, will continue to experience a decline in ecosystem services thereby affecting the general livelihoods of the communities. The natural resources and ecosystems in the landscape are also significantly damaged and will continue to be so in the future. Newbold et al. [4] showed that there could be irreversible changes to ecosystems structure in the most intensively used landscapes. This implies that the loss of multi millions of $US that related to ES impairments and natural resources degradation could reach an irreversible stage in the study landscape. Our results also identified priority areas and ecosystems (i.e., grazing land, wetlands, shrub-bush land, and forest) that need restorations and development activities for policy makers and practitioners. Significant monetary harms were also explored in the changing landscape, while human pressure is continued and increased rapidly through time. It is, therefore, expected from policy makers and practitioners to comprehend and mobilize sufficient wherewithal for landscape restoration. This spatially explicit prediction of LULC could give insight for further research to develop scenarios for alternative land use and management.

## 5. Conclusions

We set out to analyze and predict landscape transformations and estimate the values of ESs in this study. Four key observations are revealed; (i) a coupling of land transfor-

mations into agricultural land, bare land, and the built-up area in the study area have been experiencing an upward trend over time, resulting in observed reduction in natural ecosystems. These landscape transitions occurred and will continue to occur at the expense of shrub-bush-woodland, wetland, water, forest, and grassland; (ii) a large amount of reduction in the total ES values in the study periods of the analysis is recorded from the whole study landscape; (iii) degradation and loss of important habitats will continue to occur as a result of anthropogenic and environmental factors that were the major drivers and causes for substantial ES values reduction; (iv) if the existing land management and land use policy persist with business as usual, twofold of $ESV_f$ of some individual ES would be decreased. Of the total estimated 17 individual services, merely food production from provisioning services, biological control from regulating services, and pollination from supporting services were increased, whereas the remaining fourteen services decreased and will continue to decrease in the study periods. Thus, if changes in LULC continue with business as usual and human pressures on natural ecosystems are neglected without proper policy and management interventions, losses of habitats and ES trade-offs would continue to drastically harm human well-being. Therefore, proper land use policy that protect natural ecosystems, sustainable intensifications, and ecosystem restoration measures are need to be applied to address impaired non-marketing ES and balance ES trade-offs. Land management interventions are needed for the entire landscape rehabilitation through protection, afforestation, and conservation practices.

**Author Contributions:** Conceptualization, A.A.B., B.B., S.G.G., A.M.M., T.H.N., W.A., L.T. and A.E.; methodology, A.A.B., B.B., S.G.G., A.M.M., T.H.N., W.A., L.T. and A.E.; software, A.A.B., B.B., S.G.G., A.M.M., T.H.N., W.A., L.T. and A.E.; validation, A.A.B., B.B., S.G.G., A.M.M., T.H.N., W.A., L.T. and A.E.; formal analysis, A.A.B., B.B., S.G.G., A.M.M., T.H.N., W.A., L.T. and A.E.; investigation, A.A.B., B.B., S.G.G., A.M.M., T.H.N., W.A., L.T. and A.E.; resources, A.A.B., B.B., S.G.G., A.M.M., T.H.N., W.A., L.T. and A.E.; data curation, A.A.B., B.B., S.G.G., A.M.M., T.H.N., W.A., L.T. and A.E.; writing—original draft preparation, A.A.B.; writing—review and editing, A.A.B., B.B., S.G.G., A.M.M., T.H.N., W.A., L.T. and A.E.; visualization, A.A.B., B.B., S.G.G., A.M.M., T.H.N., W.A., L.T. and A.E.; supervision, B.B., S.G.G., A.M.M., T.H.N., W.A., L.T. and A.E.; project administration, A.A.B. and B.B., funding acquisition, A.A.B. All authors have read and agreed to the published version of the manuscript.

**Funding:** This research was funded by: (1) UK Research & Innovation (UKRI) through the Global Challenges Research Fund (GCRF) programme, Grant Ref: ES/P011306/ under the project Social and Environmental Trade-offs in African Agriculture (SENTINEL) led by IIED in part implemented by the Regional Universities Forum for Capacity Building in Agriculture (RU-FORUM). (2) The British Ecological Society for financing this research through Ecologists in Africa 2021 [grant number EA21/1204] complement grant for PhD study of the lead author. (3) The Eastern and Southern Africa Higher Education Center of Excellence Project: under the frame Africa Centre of Excellence for a PhD study in Climate Smart Agriculture and Biodiversity Conservation; The APC was funded by UK Research & Innovation (UKRI) through the Global Challenges Research Fund (GCRF) programme, Grant Ref: ES/P011306/ under the project Social and Environmental Trade-offs in African Agriculture (SENTINEL) led by IIED in part implemented by the Regional Universities Forum for Capacity Building in Agriculture (RUFORUM).

**Acknowledgments:** This research was supported by the CGIAR Research Program on Climate Change, Agriculture and Food Security (CCAFS) and the Global Research Alliance on Agricultural Greenhouse Gases (GRA) through their CLIFF-GRADS programme. CCAFS capability building objectives are carried out with support from CGIAR Trust Fund and through bilateral funding agreements. For details please visit https://ccafs.cgiar.org/donors (accessed on 1 March 2022). Thank you to Alliance Bioversity-CIAT, Ethiopia Office, Addis Ababa for hosting the recipient, and to the Government of New Zealand for providing financial support.

**Conflicts of Interest:** The authors declare no conflict of interest.

# Appendix A

**Table A1.** Land use land cover (LULC) transition probability matrix of 1986 and 2021; 2021 and 3031 and 2051.

| LULC | Year | Cultivated Land | Forests | Water | Grazing Land | Wetland | Built-Ups | Bare Land | Shrub-Bush Land |
|---|---|---|---|---|---|---|---|---|---|
| Cultivated land | 1986–2021 | 0.8536 | 0.001097 | 0.000744 | 0.001831 | 0.001134 | 0.008197 | 0.036208 | 0.09719 |
| | 2021–2031 | 0.992328 | 0.000356 | 0 | 0 | 0.000002 | 0.000021 | 0.000163 | 0.007129 |
| | 2021–2051 | 0.988219 | 0 | 0 | 0 | 0 | 0 | 0.004577 | 0.007205 |
| Forests | 1986–2021 | 0.165091 | 0.482923 | 0.000024 | 0.0008 | 0.000337 | 0.002785 | 0 | 0.34804 |
| | 2021–2031 | 0.1989 | 0.648039 | 0 | 0 | 0.000092 | 0.00055 | 0.001948 | 0.15047 |
| | 2021–2051 | 0.062337 | 0.664898 | 0 | 0 | 0 | 0.000167 | 0.018208 | 0.254389 |
| Water | 1986–2021 | 0.00359 | 0.0001 | 0.965713 | 0 | 0.002693 | 0.000064 | 0.019722 | 0.008119 |
| | 2021–2031 | 0.000436 | 0.000032 | 0.99913 | 0 | 0.000015 | 0.000035 | 0.000001 | 0.000351 |
| | 2021–2051 | 0.003445 | 0 | 0.992824 | 0 | 0 | 0.000055 | 0.000536 | 0.00314 |
| Grazing land | 1986–2021 | 0.736578 | 0.011584 | 0.000003 | 0.129966 | 0.000357 | 0.001856 | 0.004018 | 0.115638 |
| | 2021–2031 | 0.61181 | 0.000018 | 0 | 0.387846 | 0 | 0.000072 | 0 | 0.000254 |
| | 2021–2051 | 0.967405 | 0 | 0 | 0.032052 | 0 | 0 | 0.000054 | 0.000489 |
| Wetland | 1986–2021 | 0.134748 | 0.010659 | 0.101099 | 0 | 0.478243 | 0.000836 | 0.002355 | 0.272061 |
| | 2021–2031 | 0.02669 | 0.00037 | 0 | 0 | 0.887268 | 0.0033 | 0.000383 | 0.081988 |
| | 2021–2051 | 0.151295 | 0 | 0 | 0 | 0.830916 | 0.000096 | 0.000603 | 0.017091 |
| Built-ups | 1986–2021 | 0.081409 | 0.034978 | 0.000431 | 0 | 0.004803 | 0.691976 | 0.041505 | 0.144898 |
| | 2021–2031 | 0.129373 | 0.000297 | 0 | 0 | 0.000982 | 0.772718 | 0.000137 | 0.096493 |
| | 2021–2051 | 0.280993 | 0.000011 | 0 | 0 | 0 | 0.666244 | 0.015715 | 0.037037 |
| Bare land | 1986–2021 | 0.202829 | 0.000234 | 0.001151 | 0.000008 | 0.000209 | 0.010016 | 0.775213 | 0.010339 |
| | 2021–2031 | 0.371053 | 0.000143 | 0 | 0 | 0.000004 | 0.000101 | 0.62729 | 0.001409 |
| | 2021–2051 | 0.017684 | 0 | 0 | 0 | 0 | 0 | 0.978254 | 0.004062 |
| Shrub-bush land | 1986–2021 | 0.530804 | 0.011326 | 0.001539 | 0.001838 | 0.002232 | 0.006745 | 0.003997 | 0.44152 |
| | 2021–2031 | 0.612267 | 0.005562 | 0 | 0 | 0.000172 | 0.000551 | 0.000213 | 0.381236 |
| | 2021–2051 | 0.792051 | 0 | 0 | 0 | 0 | 0.000009 | 0.000067 | 0.207872 |

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
