# Peer review of "Ecosystem Service Valuation along Landscape Transformation in Central Ethiopia"

_land, doi:10.3390/land11040500_

Round 1

Reviewer 1 Report

The study considers past and projected changes in the area of different types of land cover in Central Ethiopia, as well as an assessment of the likely change in the value of ecosystem services. It is actual topic in the context of global changes. This work is important, but the submitted manuscript requires significant revision.

Methods (Section 2.5). 1) Authors apply the value transfer approach and use values of 17 ES from the study of Kindu et al. It would be useful to present in the article a brief information about for which territory and by what method these coefficients were obtained. 2) It is necessary to explain why the authors use for shrubs coefficients previously obtained for tropical forest (Table 2), if they define this land cover type as "land covered with herbaceous plants" (Table 1).

Presentation of results. 1) All the figures are of extremely poor quality, almost nothing is visible. 2) There are a lot of tables in the work, some of them are quite large. Their perception is difficult. Perhaps some of the results shown in the tables should be presented in the form of diagrams? 3) Some tables do not indicate in what units the values shown there are measured (Tables 6, 9, 10). 4) Some tables show the change in LULC and ES indicators as a percentage, but it is not indicated in relation to what value these percentages are calculated (Tables 4, 5, 7, 8, 9). 5) Table 10 is erroneously numbered as 9. In the text, in some places, erroneous references to table numbers are given. For details see attached PDF file.

Discussion. 1) A significant part of this section should be moved either to the "methods" section or to the "results" section (see comments in the PDF file). 2) As follows from the Table 9 (which should actually be 10), there is a significant number of studies similar to this one  for Ethiopia. However, there is no clearly presented comparison of the presented methods and results with other studies. It is only said that the results obtained by the authors "was in line with 90% studies...". What is new in this work compared to previous works? A possible reason for the discrepancy between the results and the two of the cited papers is essentially not explained. 3) In this section, it would be useful to see at least a brief discussion of possible climatic and anthropogenic drivers of land cover change (the manuscript only mentions "improper water abstaction") as well as conservation and land use policies. 

Language. There are many mistakes and misuse of words in the text. Some phrases are not clear and should be rephrased. Some of these cases are highlighted in the PDF file. The abbreviation ESV is used in the text with different meanings. 

For details see attached PDF file

Author Response

Dear Reviewer 

We are grateful to the Reviewers for the crucial and constructive comments and feedback on the manuscript. We have attempted to revise and improve our manuscript based on the reviewer’s comments and suggestions.  So, we have revised the manuscript based on all concerns that were raised from Reviewer 1 accordingly.

We have attached herewith the document with a detailed response to your comment

regards, 

Reviewer 2 Report

Comments and suggestions:

The changes of landscape and land use brought by urbanization have a profound impact on the changes of ESV, which is very important to study the changes of ESV with land use. This paper estimated the value of ESV and its changes by analyzing the landscape transformation in the past decades and predicting the future. The research results are of great significance for landscape management in the study area. In this paper, the research method was reasonable and feasible, the research data was sufficient, and the research results were reasonable. But there are still shortcomings. For example, the research method is not innovative enough, the discussion part is not deep enough and too broad, and the problems revealed based on the research results are not clear enough, leading to the inadequate application of the conclusions. Thus, major revision is suggested. Specific modification suggestions are as follows:

  • Line 31-32: It is suggested that the main conclusions related to policies and measurements should be expanded in the abstract.
  • Line 46: Why do urban areas belong to semi-natural ecosystem? Artificial ecosystems or artificial environments are more accurate.
  • Line 92-97: In the introduction, the authors pointed out the limitations of ESV method. So, can the method used in this paper break through the above limitations? Or is the level of research in this paper immune to the above drawbacks? Without a clear explanation of these issues, the limitations of ESV methods mentioned in the introduction may make people question the validity of the study. It is suggested to supplement the breakthrough of this research method in dealing with the above drawbacks; Or move them into the discussion to illustrate the limitations.
  • Line 135: In addition to data sources, it is suggested to add specific data processing methods to increase the scientific nature of the method.
  • Line 193: Why the forecast year is 2031 and 2051, skipping year of 2041 (ten years apart)? Please give the reasons.
  • Line 270 & 306: Figures accuracy are not enough, please provide HD versions.
  • Line 405: The discussion section is too general and does not further explain the possible mechanism and reason of the correlation revealed in the results. Recommendations based on results are also vague. It is suggested that the discussion be deepened. For example, “Therefore, proper policy and management interventions are needed for the entire landscape rehabilitation through protection, afforestation and management practices. ” What is proper policy and management interventions? What are the practices? It is hoped that some more can be added to improve the applicability and guidance of the results.

Author Response

Dear, Reviewer 2

We are grateful to the Reviewer for the crucial and constructive comments and feedback on the manuscript. 

We have revised the manuscript based on all comments that were raised by the reviewer. We have attached herewith our response in detail

regards,

Round 2

Reviewer 1 Report

The authors corrected all the shortcomings I indicated. Particularly impressive is the amount of work that has gone into presenting tabular data as charts. The manuscript can now be published

Author Response

Dear, Reviewer

Thank you so much for your comments. 

Kindly, find the revised manuscript attached herewith 

regards, 

Reviewer 2 Report

Comments and suggestions:

After the previous round of major revision, the authors made better modifications and responses to the suggestions: specific relevant policies and measures were added to the abstract; the expression of "semi-natural ecosystems" was corrected as "modified/artificial ecosystems", which was more in line with its classification; added a discussion of the limitations of the ESV method; the data processing method was supplemented; and finally, a certain degree of refinement of results-based applications in the discussion section was provided, although I think it is not specific and grounded enough, but it has provided more inspiration than the previous version. On this basis, there is also remained a comment and suggestion: “Simulations were employed in ten years time interval and then 2031 and 2051 were selected as an optimum years that prompt for the mid and long term prediction. ” (Lines 229-231) – Although the authors added the above statement, they still did not explain why years of 2031 and 2051 is the "optimum year". What are the characteristics or particularities that make them the year of concern for this study? It can be tried to interpret from the perspective of the characteristics, current situation and development plan of the study area.

Author Response

Dear, Reviewer

Thank you so much for your constructive comments. 

Kindly, find the attached revised manuscript herewith.

regards, 
